# Genome-Wide Association Analysis of Plant Height Using the Maize F1 Population

**DOI:** 10.3390/plants8100432

**Published:** 2019-10-21

**Authors:** Yong Zhang, Jiyu Wan, Lian He, Hai Lan, Lujiang Li

**Affiliations:** Maize Research Institute, Sichuan Agricultural University/Key Laboratory of Crop Genetic Resources and Improvement, Ministry of Education, Chengdu 611130, Sichuan, China; YongZhang978812203@163.com (Y.Z.); wanjiyu61@163.com (J.W.); helian912@126.com (L.H.); lanhai_maize@163.com (H.L.)

**Keywords:** plant height, GWAS, F1, maize

## Abstract

Drastic changes in plant height (PH) are observed when maize adapt to a higher plant density. Most importantly, PH is an important factor affecting maize yield. Although the genetic basis of PH has been extensively studied using different populations during the past decades, genetic basis remains unclear in the F1 population, which was a widely used population in production. In this study, a genome-wide association study (GWAS) was conducted using an F1 population consisting of 300 maize hybrids with 17,652 single nucleotide polymorphisms (SNPs) makers to identify candidate genes for controlling PH. A total of nine significant SNPs makers and two candidate genes were identified for PH. The candidate genes, Zm00001d018617 and Zm00001d023659, were the genes most probable to be involved in the development of PH. Our results provide new insights into the genetic basis of PH in maize.

## 1. Introduction

Maize (*Zea mays* L.) is a major worldwide crop, as well as the main food source for people in developing countries. Thus, constant improvement of maize yield is an important long-term target of breeders. During the past several decades, maize yield has approximately increased eightfold [1]. The large increase of maize yield was not a result of advanced plant technology and modernized management, but rather it was due to hybrid breeding and the increase of plant density [1,2]. In maize, plant height (PH) is an extremely important agronomic trait easily measured in the field. However, resulting from selection by breeders, PH has significantly changed for adaptation of a higher plant density. Excessive PH may lead to plant lodging, resulting in the decreasing of yield. A reasonable PH is beneficial for increasing plant density and maize yield. Therefore, dissecting the genetic basis of PH would provide a theoretical direction for maize breeding.

Genome-wide association study (GWAS) has been widely used in the study of dissecting complex quantitative traits. PH is a complex quantitative trait that is controlled by multi-genes and easily affected by the environment. Currently, numerous quantitative trait loci (QTLs) and candidate genes related to PH in maize have been identified. For example, He, et al. [3] identified 10 QTLs for PH in 150 recombinant inbred line (RIL) populations under six environments. Lan, et al. [4] identified 6 QTLs for PH in an F2 population consisting of 191 individuals. Li, et al. [5] detected 7 QTLs for PH in 161 DH lines. Liu, et al. [6] identified 6 candidate genes for PH in 284 RIL populations. Vanous, et al. [7] identified 18 candidate genes related to height in 252 DH lines. Several genes related to PH have been cloned and undergone verification, such as *d1*, *d2*, *d3*, *d5*, *d8*, *d9*, *An1*, *DWF1*, *DWF4*, and *ZmGA3ox2* [8,9,10,11,12,13,14,15,16,17]. Although the PH has been extensively studied through different maize populations, the genetic architecture remains unclear in the F1 population.

In our study, we used the F1 population as the associated population to dissect the genetic basis of PH and identify candidate genes for PH in multiple environments by GWAS. The F1 population, consisting of 300 hybrids, was constructed through incomplete diallel crosses using 99 maize inbred lines genotyped by genotyping-by-sequencing (GBS) technology [18], which is an improved strategy for addressing a complex genome via next-generation sequencing technology. PH was measured in five environments for one year. Candidate genes significantly associated with PH were further screened and analyzed. Our study aimed to improve understanding of the genetic basis of PH and provide new insights into the genetic basis of PH in maize.

## 2. Materials and Methods

### 2.1. Materials and Field Trials

The F1 population in this study consisted of 300 hybrids that were derived from 99 maize inbred lines through an anomalous diallel crosses method. These hybrids were planted with two replicates per environment based on a randomized complete block design. Each hybrid was grown in a single row. Each plot contained a single row (10 plants, two plants each pit) 2 m in length and 0.80 m from the next row; the planting density was approximately 60,000 individuals/ha. PH was measured in five environments in 2017, including Yunnan (Jinghong, 100°46′ E, 22°0′ N), Sichuan (Mianyang, 104°44′ E, 31°28′ N), Sichuan (Ya’an, 103°0′ E, 29°59′ N), Sichuan (Luding, 102°14′ E, 29°54′ N), Sichuan (Yibin, 104°37′ E, 28°45′ N). PH, which was measured as the distance in centimeters from the soil level to the tassel at reproductive maturity, was evaluated in each hybrid by randomly selecting five plants. The trait value for each hybrid was averaged for the measured plants. The field trial was conducted for conventional field management.

### 2.2. Analysis of Phenotypic Data

Analysis of variance and descriptive statistical analysis of the phenotypic data for PH were performed by IBM SPSS statistics version 20.0 software. The broad-sense heritability (H^2^) of trait was estimated as described by Knapp [19] as H^2^ = V_G_/(V_G_ + V_GE_/*n* + V_residual_/*rn*), where V_G_ is the genotypic variance, V_GE_ is the interaction variance of genotype × environment, V_residual_ is the error variance, *n* is the number of environments, and *r* is the number of replications per environment. The R package “lme4” was used for computing the best linear unbiased prediction (BLUP) values under five environments [20].

### 2.3. Analysis of Genotypic Data

The genotypic data of 99 inbred lines were obtained by genotyping-by-sequencing (GBS) technology [18] and contained 559,678 SNPs makers. SNPs with heterozygous or missing were expurgated; only SNPs markers that were homozygous in 99 inbred lines were reserved. Finally, 29,328 SNPs markers were obtained and used for inferring genotype of the hybrids, which was generated according to the genotype of their parents referenced to Zhao, et al. [21]. SNPs with a minor allele frequency (MAF) <0.05 were expurgated, and only biallelic sites were reserved. The resulting 17,652 SNPs were subsequently used for population structure analysis, kinship coefficients analysis, LD calculation, and GWAS analysis. The population structure(Q matrix) was estimated by using the STRUCTURE 2.3.4 software program [22]. The number of subgroups (K) was set from 1 to 16, burn-ins were set to 10,000, and Markov Chain Monte Carlo (MCMC) was set to 10,000 and conducted using the mixture model and correlated allele frequency for each K. Based on the output log likelihood of data (LnP(D)) of STRUCTURE, the ad hoc statistic ∆K was applied to determine the reasonable subgroups number [23]. Kinship (K matrix) and LD were estimated by using TASSEL 5.0 software program. Microsoft Excel 2016 software program was used to create an LD attenuation plot and kinship map. The physical distance corresponding to an R2 value of 0.2 was assumed as the distance of LD attenuation.

### 2.4. Genome-Wide Association Study

The FarmCPU model, a fixed and random effect model in which population structure(PCA) is a fixed effect and kinship is a random effect in this model, was adapted for GWAS [24]. SNPs maker with a *p* value of less than threshold P_threshold_ = 0.05/N, where N was the total number of SNPs makers, was considered to be significantly associated with the trait [25]. The threshold level for marker-trait significant associations was set as 2.83 × 10^−6^ in this study. The whole genome sequence of B73 was used as a reference genome for identifying candidate genes [26,27]. Based on the physical positions of the SNPs, which were significantly associated with target traits, the maizeGDB database (B73_RefGen_v4) was used for identifying candidate genes and the functional annotations. Herein, we searched the candidate genes according to the region of LD decay (the average LD distance among 10 chromosomes was 200 kb).

## 3. Result

### 3.1. Phenotypic Analysis and Heritability

Descriptive statistical analysis of PH BLUP phenotype data is shown in Table 1. PH ranged from 239.15 cm to 324.50 cm, with an average of 290.67 cm and a coefficient of variation of 5.62%. Skewness and kurtosis of PH were between −1 and 1, indicating that the phenotype conformed to a normal distribution (Figure 1). Analysis of variance showed that the phenotype of PH was strongly affected by genotype, environment, and genotype×environment interactions (Table 2). The broad-sense heritability (H^2^) of PH was more than 80%, indicating that PH was mainly controlled by genetics. The above results indicated that the measured phenotype is reliable for dissecting the genetic basis of PH.

### 3.2. LD Analysis

A total of 17,652 high-quality SNPs makers were used to evaluate genome-wide LD attenuation through the TASSEL 5.0 software program. The LD attenuation plot is shown in Figure 2. The LD attenuation plot indicated that LD decreased rapidly as the genetic distance increasing, and the attenuation rate for each chromosome was different. The attenuation rate of chromosome 2 (100 kb) was the fastest, and the attenuation rate of chromosome 3 (300 kb) was the slowest. The average LD attenuation distance among 10 chromosomes was 200 kb (R^2^ = 0.2, Figure 2).

### 3.3. Population Structure Analysis

Based on 17,652 high-quality SNPs markers, the population structure of 300 hybrids was divided by the STRUCTURE 2.3.4 software program. The reasonable subgroups number referred to the method proposed by Evanno, et al. in 2005 [28]. Analysis results showed that ∆K was the largest value when K = 4, indicating that the 300 hybrids were most suitable for being divided into four subgroups (Figure 3).

### 3.4. Kinship Analysis

Based on 17,652 high-quality SNPs markers, the kinship of 300 hybrids in this study was evaluated by the TASSEL 5.0 software program. The analysis results suggested that kinship of 96.4% of the hybrids was between 0 and 0.5, and kinship of 3.6% of the hybrids was between 0.5 and 1 (Figure 4). On the whole, hybrids of this study had a broad genetic basis.

### 3.5. GWAS Analysis

The BLUP phenotype data of PH and genotype data of 17,652 high-quality SNPs markers were analyzed for GWAS by FarmCPU model. The Q matrix (PCA) and K matrix (Kinship) were fitted into the FarmCPU model for reducing false positives. Quantile quantile (Q-Q) plots indicated that the population structure and kinship were well controlled for GWAS in the FarmCPU model (Figure 5). Manhattan plots showed that nine SNPs makers were significantly associated with PH (Figure 5). SNP markers, which were significantly associated with PH, were located on chromosomes 1, 2, 4, 7, 9, and 10 (Table 3). According to nine significant SNPs makers, two candidate genes related to PH were identified in this study. The details of candidate genes and the functional annotations are shown in Table 4. The functions of the two candidate genes were related to plant hormones.

## 4. Discussion

Height is regarded as one of the most important agronomic traits in maize and has a strong effect on yield, planting density, and lodging resistance. However, the potential genetic mechanism remains unclear. In this study, the phenotypic data of PH conformed to a normal distribution, suggesting that this trait is a quantitative trait controlled by multi-genes with low effect. In addition, PH had a high heritability of more than 80%. All findings were consistent with those of previous studies [7,29], indicating that PH was mainly controlled by genetics and is suitable for early generation selection.

Various types of plant populations have been used for studying the complex quantitative traits, including bi-parental and multi-parental populations and natural populations. Natural populations that consisted of the inbred lines in maize have been the most widely used for genome-wide association analysis (GWAS) of complex quantitative traits. Indeed, the analysis result of GWAS based on the inbred lines cannot directly be used to dissect the genetic basis in the F1 population. In this study, the F1 population consisting of 300 hybrids derived from 99 maize inbred lines was used for GWAS. Compared to the inbred lines population, the F1 population adapted better to different environments. As a result, the F1 population provided the possibility for testing across multi-environments. At present, the cost of sequencing remains very high, but using the F1 population can reduce the cost of genotyping. As only the parental inbred lines be genotyped, the genotype of the hybrids could be inferred by bi-parental genotype [21]. For example, 100 inbred lines can be used to generate [(100 − 1) × 100]/2 = 4500 hybrids. It would be more expensive to genotype 4500 hybrids compared to 100 inbred lines. However, the genotypes of 4500 hybrids can be inferred through the genotypes of 100 parental inbred lines. Therefore, the cost of genotyping would be greatly reduced.

Currently, GWAS has become an efficient and powerful tool for identifying the candidate genes related to the quantitative trait. In our study, a total of two candidate genes were identified for PH according to the physical location of nine significantly associated SNPs markers. However, genes previously identified did not overlap with the candidate genes in this study. A possible reason for this finding is that GWAS is relatively insensitive to detecting low-frequency loci with significant effects, although the low-frequency or rare allelic variants seem to be important for the target trait [21].

The detection of dwarf and semi-dwarf mutants provided a possibility of dissecting the genetic basis of PH. A few genes controlling PH have been identified using dwarf and semi-dwarf mutants. In semi-dwarf rice mutants, semi-dwarf phenotype controlled by *sd1* results from a deficiency of GA20-oxidase activity in the gibberellin (GA) biosynthetic pathway [30]. In wheat, a semi-dwarf phenotype was controlled by the *RHT* gene, which encodes a DELLA protein involved in the GA signaling [31]. In maize, a few genes related to PH have been cloned. For example, *Dwarf3* (*D3*) encodes a cytochrome P450 involved in the initial phase of GA biosynthesis [12]. *ZmGA3ox2,* which modified the PH by approximately 20 cm and is a candidate gene for a major QTL qPH3.1, encodes a GA3 β-hydroxylase, an important enzyme in the GA biosynthesis path [17]. These previous studies have shown that GA biosynthesis and signaling played a key role in the development of PH [32]. In addition, auxin (IAA) synthesis and signaling also played a key role in the development of PH. A multidrug resistant-like ABC transporter involved in polar auxin transport reduced plant height in the maize *br2* mutant [33]. In this study, the candidate gene Zm00001d018617, which encodes gibberellin 2-oxidase 12, involved GA biosynthesis, and the candidate gene Zm00001d023659, which encodes auxin response factor 2, was involved in IAA biosynthesis. Therefore, the candidate genes Zm00001d018617 and Zm00001d023659 were most probably potential genes involved in the development of PH.

In summary, the candidate genes for PH were identified on chromosomes 7 and 10. Two candidate genes were related to the plant hormones GA and IAA.

## 5. Conclusions

In this study, the F1 population consisting of 300 hybrids was used for GWAS using the FarmCPU model, and 9 significant SNPs makers associated with PH were detected. According to the physical location of significant SNPs makers, two candidate genes were identified. The candidate genes, Zm00001d018617 and Zm00001d023659, were the genes most probable to be involved in the development of PH. The two candidate genes will be the subject of next study.

## Figures and Tables

**Figure 1 plants-08-00432-f001:**
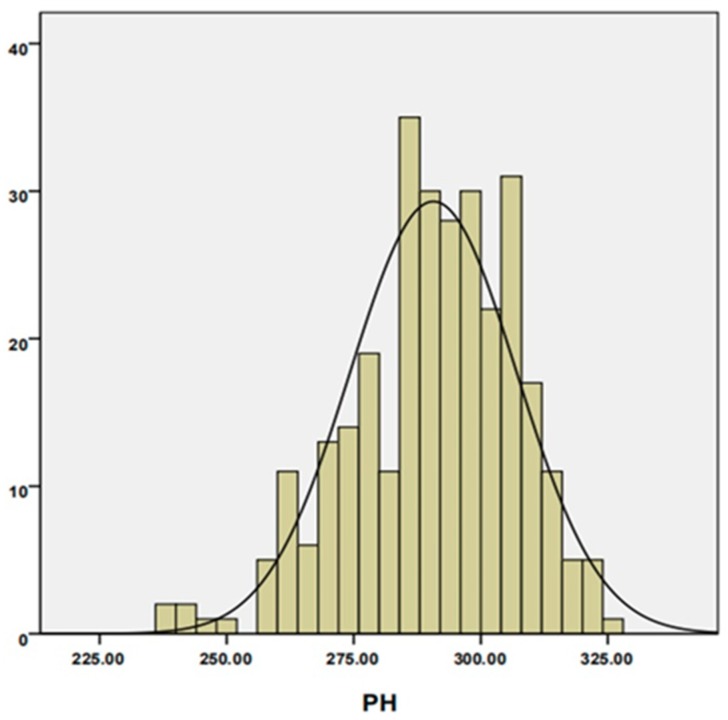
The phenotype distribution of PH.

**Figure 2 plants-08-00432-f002:**
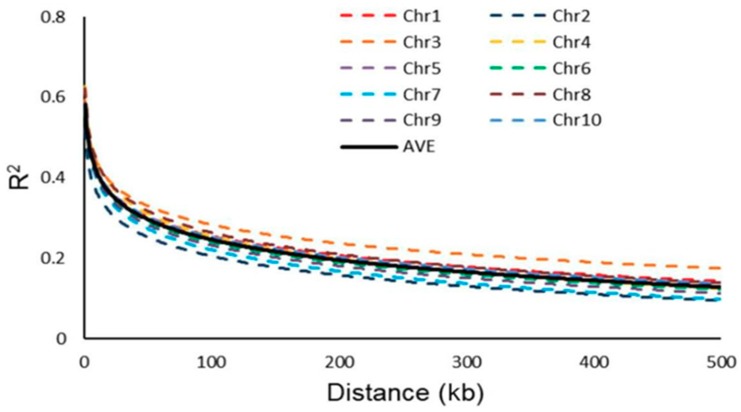
LD decay rate per chromosome based on mean R^2^ per 100 kb region.

**Figure 3 plants-08-00432-f003:**
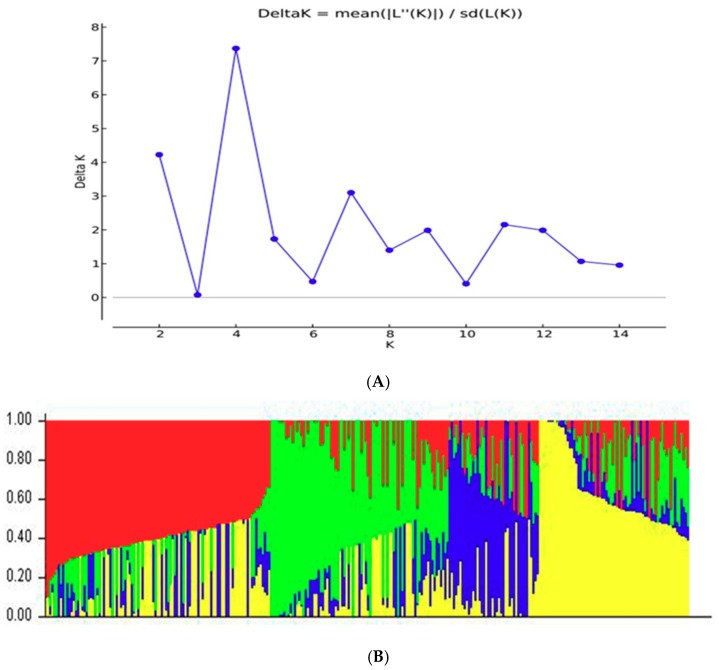
Analysis of the population structure of the 300 hybrids estimated from 17,652 SNPs. (**A**) ∆K value related to different K; (**B**) Population structure of the 300 hybrids from K = 4.

**Figure 4 plants-08-00432-f004:**
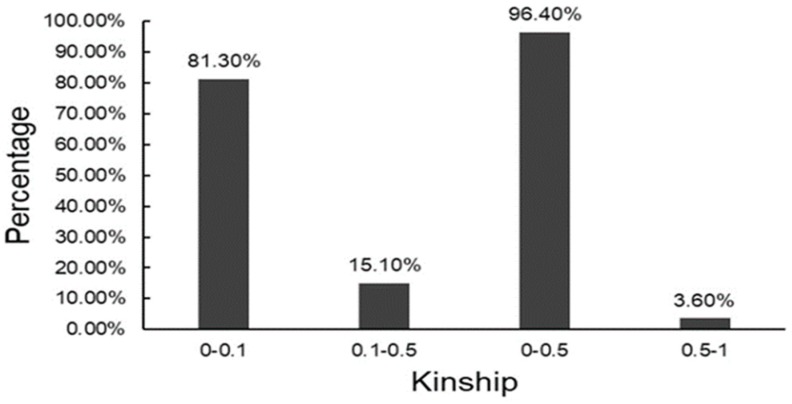
The distributions of kinship between 300 hybrids.

**Figure 5 plants-08-00432-f005:**
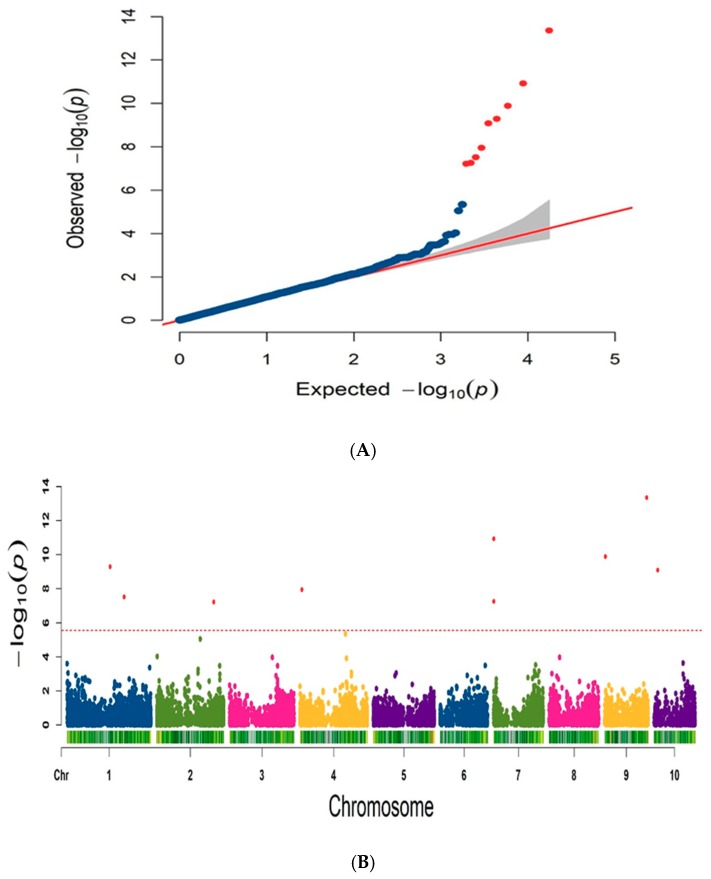
Q-Q plot and Manhattan plot of the significant association detected for PH. (**A**) Q-Q plot; (**B**) Manhattan plot.

**Table 1 plants-08-00432-t001:** Descriptive statistical analysis and the broad-sense heritability (H^2^) of PH. SD = standard deviation, CV = coefficient of variation, H^2^ = The broad-sense heritability.

Trait	Mean	Minimum	Maximum	SD	Skewness	Kurtosis	CV (%)	H^2^ (%)
PH	290.67	239.15	324.50	16.34	−0.56	0.26	5.62	83.32

**Table 2 plants-08-00432-t002:** Analysis of variance for PH. SS = sum of squares, DF = degree of freedom, MS = mean squares and Sig = significance, G = genotype, E = environment, G × E = genotype × environment, **: indicates significance at level of 0.01.

Trait	Source	SS	DF	MS	F Value	Sig
PH	G	954,834.54	299	3193.43	32.98	<0.01 **
	E	1,122,082.30	4	280,520.57	2897.20	<0.01 **
	G×E	363,150.96	1194	304.15	3.14	<0.01 **
	Error	142,235.55	1469	96.83		

**Table 3 plants-08-00432-t003:** Summary of the significant makers for PH.

Trait	Chr	SNP ID	SNP Physical Position	*p* Value
PH	1	SNP−41465	157,566,180	5.19 × 10^−10^
	1	SNP−54654	208,218,658	3.05 × 10^−8^
	2	SNP−137325	207,979,043	6.08 × 10^−8^
	4	SNP−214914	7,407,530	1.14 × 10^−8^
	7	SNP−382338	1,148,863	5.57 × 10^−8^
	7	SNP−382339	1,148,876	1.20× 10^−11^
	9	SNP−480424	3,878,215	1.31 × 10^−10^
	9	SNP−516792	154,433,535	4.49 × 10^−14^
	10	SNP−524015	13,705,944	8.83 × 10^−10^

**Table 4 plants-08-00432-t004:** The functional annotations of candidate genes identified in this study.

Trait	Chr	SNP ID	Gen ID	Encoding
PH	7	SNP−382338	Zm00001d018617	gibberellin 2–oxidase 12
	10	SNP−524015	Zm00001d023659	auxin response factor 2

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
