# Peer review of "Genome-Wide Association Analysis of Plant Height Using the Maize F1 Population"

_plants, 2019, doi:10.3390/plants8100432_

Round 1
Reviewer 1 Report
Dear Author
Please read carefully and correct typos and grammatical mistakes.
e.g. line 134 "3.4. Kingship analysis" correct as "Kinship analysis"
Best
Reviewer 2 Report
The topic of the article is topical and significantly supplements the existing scientific information on the trait being studied. Appropriate and modern methods have been selected for the research, which ensure reliable results. The results are clearly presented and discussed in details.
Reviewer 3 Report
The manuscript examines the interest of dissecting association between molecular markers and plant height in corn using a GWAS approach.
The work is interesting and the conclusions could have
favorable uses for corn breeding.
The approach is clear, the experimentation is simple, and the
molecular and statistical methods used are sounds. The discussion is the most
interesting section of this manuscript as it presents contradictory results and
evokes the limits of the used methods.
However, there are things that are very vague and necessarily
require clarification.
1 - In the introduction, the authors discuss the interest of
selection on the basis of the plant height to improve yield without explaining
the link between these two characters. It is therefore important to understand
the process and for all readers that this is made clear in the introduction.
2- the objective of this study needs to be clarified.
P1 L43-47. The authors quote "in our study ... (GBS)
technology [18]. This is unclear and needs to be rewritten differently.
Moreover, the originality of this work is not demonstrated compared to other
studies [7-18].
3- The conclusion of this work, despite the interesting size
of F1 examined, must be moderate. Indeed, this work highlights a little the
limit of the method used for the detection of loci at low frequencies. Genetic
funds also diverge for these low-frequency loci. Thus, what was detected in
this study, is not necessarily applicable in other studies.
Additional remarks
P1 L27 please add plant height again and put PH between brackets.
P1 L35-39. In order to avoid confusion,
Please cite the number of reference just after citing the authors.
For example “He et al. [3]” instead of He et al. identified…..under six environments [3].
